# Mitochondrial Dysfunction, Oxidative Stress, and Inter-Organ Miscommunications in T2D Progression

**DOI:** 10.3390/ijms25031504

**Published:** 2024-01-25

**Authors:** Rajakrishnan Veluthakal, Diana Esparza, Joseph M. Hoolachan, Rekha Balakrishnan, Miwon Ahn, Eunjin Oh, Chathurani S. Jayasena, Debbie C. Thurmond

**Affiliations:** Department of Molecular and Cellular Endocrinology, Arthur Riggs Diabetes & Metabolism Research Institute, City of Hope Beckman Research Institute, 1500 E. Duarte Rd, Duarte, CA 91010, USA; desparza@coh.org (D.E.); jhoolachan@coh.org (J.M.H.); rbalakrishnan@coh.org (R.B.); mahn@coh.org (M.A.); euoh@coh.org (E.O.); cjayasena@coh.org (C.S.J.)

**Keywords:** oxidative stress, insulin resistance, islet beta cells, type 2 diabetes, extracellular vesicles, mitochondria, prediabetes

## Abstract

Type 2 diabetes (T2D) is a heterogenous disease, and conventionally, peripheral insulin resistance (IR) was thought to precede islet β-cell dysfunction, promoting progression from prediabetes to T2D. New evidence suggests that T2D-lean individuals experience early β-cell dysfunction without significant IR. Regardless of the primary event (i.e., IR vs. β-cell dysfunction) that contributes to dysglycemia, significant early-onset oxidative damage and mitochondrial dysfunction in multiple metabolic tissues may be a driver of T2D onset and progression. Oxidative stress, defined as the generation of reactive oxygen species (ROS), is mediated by hyperglycemia alone or in combination with lipids. Physiological oxidative stress promotes inter-tissue communication, while pathological oxidative stress promotes inter-tissue mis-communication, and new evidence suggests that this is mediated via extracellular vesicles (EVs), including mitochondria containing EVs. Under metabolic-related stress conditions, EV-mediated cross-talk between β-cells and skeletal muscle likely trigger mitochondrial anomalies leading to prediabetes and T2D. This article reviews the underlying molecular mechanisms in ROS-related pathogenesis of prediabetes, including mitophagy and mitochondrial dynamics due to oxidative stress. Further, this review will describe the potential of various therapeutic avenues for attenuating oxidative damage, reversing prediabetes and preventing progression to T2D.

## 1. Introduction

### 1.1. Prediabetes and Type 2 Diabetes

Over the last decade, type 2 diabetes (T2D) has emerged as a complex multifactorial heterogenous disease [1]. About 90–95% of all diabetes cases worldwide are composed of T2D cases [2,3]. Individuals are clinically diagnosed with T2D when there is a relative insulin deficiency (i.e., islet β-cell dysfunction) combined with peripheral insulin resistance (IR) [3]. Alarmingly, 541 million adults worldwide are currently living with prediabetes, a transitional asymptomatic hyperglycemic state between normoglycemia and T2D, and these individuals are at a significant risk of developing T2D and other metabolic complications that impact their lifespan and quality of life [2]. Four T2D archetype profiles in newly diagnosed individuals were recently identified based on differences in disease progression, genetic risk scores, and circulating omics biomarkers (Figure 1) and they were: (1) lean and insulin deficient, (2) obese and insulin sensitive, (3) obese and IR, and (4) a severe phenotype with obesity, IR, dyslipidemia, and impaired β-cell glucose sensitivity that progresses rapidly to frank T2D that required the most anti-diabetic interventions [4].

IR is a hallmark of prediabetes and is characterized by an impaired cellular response to insulin stimulation at the peripheral tissues, such as skeletal muscle and adipose tissue [3]. This leads to dysglycemia (i.e., impaired fasting glucose and/or glucose tolerance) and metabolic dyshomeostasis. Conventionally, skeletal muscle IR is thought to precede islet β-cell dysfunction and be the primary event driving dysglycemia toward T2D. While this may be the case in obesity-related T2D, this notion is challenged by the observation that various populations have varying degrees of insulin secretory capacity and IR [5]. Critically, there is a high incidence of prediabetes and T2D in lean non-white populations with a BMI index <25 kg/m^2^ and these individuals present with insulin deficiency, rather than IR [6,7,8,9]. It is noteworthy however that the BMI may not realistically capture adiposity.

Regardless of the primary event (i.e., IR vs. β-cell dysfunction) driving dysglycemia, compelling mounting evidence suggests that prolonged pathological oxidative stress within metabolic tissues (e.g., skeletal muscle, adipose tissue, pancreatic islet β-cells) may be a crucial contributor to the metabolic abnormalities present during prediabetes and drive progression to T2D [10,11,12,13,14,15,16,17]. Thus, gaining a deeper understanding of the mechanisms that drive pathological oxidative stress could aid in the development of novel therapeutic strategies that halt/prevent progression to T2D.

### 1.2. Oxidative Stress

Oxidative stress is defined as an impairment of the defensive antioxidant systems and an elevation of the reactive oxygen species (ROS), which include superoxide anion radical (O_2_^•−^), hydrogen peroxide (H_2_O_2_) and hydroxyl radical (OH^•^), as well as reactive nitrogen radicals (RNS) and non-radical peroxynitrite (ONOO^–^) [18]. Compartmentalized and controlled ROS production has emerged as a physiological mediator of many cellular responses [19,20], including glucose-stimulated insulin secretion (GSIS) by islet β-cells and glucose-uptake related to skeletal muscle function [21,22,23,24,25].

Mitochondrial ROS (mROS) and extramitochondrial sources contribute to ROS production and there is the potential for these various sources to interact with one another to alter redox balance and promote oxidative stress [26,27]. It is well-known that glycolytic and oxidative metabolism of glucose, as well as fatty acid oxidation leads to adenosine triphosphate (ATP) generation in the mitochondria [28,29,30]. The major source of ROS generation is mitochondrial respiration during ATP production; electrons that escape the electron transport chain (ETC) reacts with oxygen to produce O_2_^•−^ and H_2_O_2_ (reviewed in [31]). Nicotinamide adenosine dinucleotide phosphate oxidases (NOXs), a membrane-associated protein complex, also promotes the generation of O_2_^•−^ using oxidation of nicotinamide adenosine dinucleotide phosphate (NADPH). In addition to the expression and localization to the cell membrane, NOX proteins also reside in the endoplasmic reticulum (ER) and in the mitochondria [32,33,34]. Peroxisomes that are armed with oxidases constitute another source of H_2_O_2_ production and are essential for lipid metabolism.

ROS is a powerful oxidant that damages many biological molecules [35,36] (Figure 2). In addition to being a significant source of cellular ROS, mitochondria are targets of cellular ROS damage. Mitochondrial DNA (mtDNA), protein, and membrane associated lipids are susceptible to oxidative stress leading to mitochondrial dysfunction [37]. Mitochondrial dysfunction is characterized by diminished ATP production, altered mitochondrial enzymatic activity, impaired redox balance and excessive generation of ROS. Thus, mitochondrial dysfunction has the capacity to further enhance oxidative stress [38,39].

Strict maintenance of glycemic control is necessary for maintaining mitochondrial function and morphology in humans and animals [40,41,42]. Mitochondrial metabolism is reduced in obese-IR individuals [43]. Furthermore, T2D or IR individuals show a deficiency for the mitochondrial biogenesis master regulator, peroxisome proliferator-activated receptor coactivator-1 alpha (PGC1α) in their skeletal muscle [44,45]. Plasma ROS production was also recently found to be significantly elevated in individuals with prediabetes and those with newly diagnosed T2D [46]. However, transient ROS elevations in skeletal muscle and islet β-cells are thought to be beneficial, adaptive and necessary in health. The threshold for ROS elevations and associated mitochondrial dysfunction that is considered detrimental to the specialized functions of skeletal muscle and pancreatic islet β-cells that lead to metabolic dyshomeostasis during prediabetes and progression to T2D remains to be established. This review will highlight new findings related to the underlying molecular mechanisms in ROS-related pathogenesis of (pre)diabetes, including mitochondrial dysfunction and inter-organ miscommunication that impacts mitochondrial function due to pathological oxidative stress. We have also attempted to organize information in this review as a continuum and present a broader picture on the contributory factors that promote T2D development including IR and β-cell dysfunction. Further, this review will describe the potential of therapeutic avenues for attenuating oxidative damage, reversing prediabetes and preventing progression to T2D.

## 2. Physiological ROS Production and Glucose Metabolism in Health

There is emerging appreciation that transient bursts of ROS from mitochondrial and extramitochondrial sources facilitates insulin release in response to glucose from islet β-cells and promotes skeletal muscle glucose uptake in health. Gaining an understanding of the various cellular ROS sources that contribute to glucose metabolism in health is essential because the dysregulation of any of these cellular ROS pathways has the potential to promote mitochondrial dysfunction and progression to T2D.

### 2.1. Glucose Sensing, Insulin Secretion and Islet β-Cells

Insulin is a hormone that is essential for whole-body glucose homeostasis and is released by islet β-cells in response to glucose. ROS is emerging as a signaling component necessary for islet β-cell glucose sensing and insulin secretion (Figure 3). The presence of glucose stimulates H_2_O_2_ generation [22,47] and has been shown to alter the intracellular redox status [22]. Further, an increase in intracellular H_2_O_2_ leads to an increase in insulin secretion [21]. Suppression of glucose-induced H_2_O_2_ accumulation by antioxidants hinders GSIS [21], and ROS activates the ryanodine receptor, which subsequently provides the necessary increase in intracellular Ca^2+^ for insulin release [48]. Islet β-cells express cell membrane-localized NOX2 [49], and the generation of ROS by guanine nucleotide exchange factor cytohesin-2 (ARNO) signaling promotes NOX2 complex assembly and GSIS in rat INS 832/13 clonal β-cells [50].

Mitochondrial blockers were shown to enhance mROS levels and elevate insulin release similar to levels stimulated by glucose [51]. Knockdown of the mitochondrial uncoupling protein 2 (UCP2) increased GSIS from INS-1E cells [52], and this was ablated by exposure to the antioxidant MnTMPyP [53]. NOX4 is reported to reside within the mitochondria and is able to generate both O_2_^•−^ and H_2_O_2_ [54]. β-cell–specific NOX4 knockout mice (NOX4βKO) and from NOX4-silenced or catalase-overexpressing INS-1E clonal β-cell studies showed that the first phase of insulin secretion was completely abolished [22]. Furthermore, lentiviral NOX4 overexpression or enhancing H_2_O_2_ rescued GSIS from NOX4βKO mice [22]. NOX4βKO mice exhibited impaired glucose tolerance and peripheral IR [22], thus, demonstrating the in vivo role of NOX4 derived ROS in insulin release. In contrast, using a pharmacological NOX4 inhibitor (GLX7013114) in human islets and human EndoC-βH1 cells showed maximal oxygen consumption rates were increased in islets after acute NOX4 inhibition [55]. In EndoC-βH1 cells, NOX4 inhibition increased the mitochondrial membrane potential, mROS and the ATP/ADP ratio. Furthermore, the insulin release from EndoC-βH1 cells at a high glucose concentration increased with NOX4 inhibition, thus suggesting that pharmacological alleviation of NOX4 could result in enhanced insulin release. However, the discrepancies in the studies may be due to small changes in ROS levels at different microenvironments that escape detection, unless a specific probe is utilized to explicitly target a particular microenvironment to study the regulation of ROS from that compartment.

In the past several decades, the islet biology field has looked at conventional antioxidant systems to protect islet β-cells from oxidative stress. New emerging evidence suggests that there are several other alternate antioxidant systems conferring protection to β-cells, such as the peroxiredoxins (PRX), thioredoxins (TRX), thioredoxin reductases (TRXR), and NRF2 [56,57]. NRF2 is implicated as a master regulator of several antioxidant genes based on its ability to confer β-cell protection against experimentally-induced supraphysiological concentrations of ROS [58]. The varying degree with which these different antioxidants offer protection against oxidative stress may depend on the generation of ROS at different subcellular microdomains and thus escape detection due to an inability to measure such changes due to their highly reactive nature.

### 2.2. Glucose Uptake and Skeletal Muscle Function

In the post-prandial state in healthy individuals, skeletal muscle accounts for ~70–90% of glucose uptake from the blood [59,60]. Transient levels of ROS produced in a controlled fashion play crucial roles in skeletal muscle cells, including the control of gene expression, regulation of cell signaling pathways, and modulation of skeletal muscle force production (reviewed in [61]). For example: (i) rat extensor digitorum longus muscles increased glucose uptake when exposed to transient elevated oxidative stress, and catalase exposure blunted glucose uptake [62]; (ii) H_2_O_2_-stimulated glucose uptake was completely inhibited by the phosphatidylinositol 3-kinase (PI3K) inhibitor wortmannin, but not the nitric oxide inhibitor NG-monomethyl-l-arginine [62]; (iii) ROS and NO have been implicated in skeletal muscle glucose uptake, via an AMP-activated protein kinase (AMPK) independent mechanism during contraction [23]. Moreover, NOX2-derived cytosolic ROS is required for insulin-stimulated glucose transporter type 4 (GLUT4)-mediated glucose uptake in skeletal muscle [24]. Relatedly, exercise-induced GLUT4 translocation and glucose uptake are impaired in knockout mouse models of NOX2 assembly components, Rac1 and p47*^phox^* [24]. The requirement for skeletal muscle NOX4 in adaptive responses that promote insulin sensitivity and maintain redox balance was recently reported [25]. Further, stabilization of the NFE2L2-mediated antioxidant defense by NOX4-derived H_2_O_2_ was necessary for maintaining insulin sensitivity in a cell autonomous manner [25]. Collectively, these findings support the notion that transient levels of ROS are important for skeletal muscle function.

## 3. Impaired Mitochondrial Dynamics and Mitophagy in Skeletal Muscle Insulin Resistance

The mitochondrial networks across the skeletal muscle fiber types undergo dynamic and heterogeneous alterations and are vital components for muscle plasticity [63] that permit adaptations to the fluctuations in skeletal muscle energy demands [64]. The progressive decline in skeletal muscle insulin sensitivity, culminating in chronic IR, is a major contributor to T2D onset. Abnormal skeletal muscle mROS levels are associated with high-fat and high-glucose diet-induced IR in humans and in preclinical models [25,65,66].

### 3.1. Impaired Mitochondrial Dynamics in Skeletal Muscle IR

Mitochondrial dynamics consists of the coordinated cycling between mitochondrial fission (division of one mitochondrion into two daughter mitochondria) and fusion (the union of two mitochondria into one mitochondrion). Typically, mitochondrial fragmentation phenotypes due to elevated mitochondrial fission activity are linked to mitochondrial dysfunction and are observed during elevated stress levels and cell death (reviewed in [67]) [68,69]. Clinical evidence from individuals with T2D supports the concept that progressive IR is associated with elevated levels of fragmented mitochondrial networks and diminished oxidative capacity [70].

A high-fat diet (HFD) increases the risk of skeletal muscle IR via increased intramyocellular lipid accumulation [64] increased ROS production [71] mitochondrial damage [72], muscle inflammation [73,74] and dysregulation of insulin signaling [75]. HFD-fed mice [76], as well as palmitate-treated human and rodent myotubes [77,78], showed increased mitochondrial fragmentation driven by the activation of the fission marker dynamin-related protein 1 (Drp1). HFD and/or palmitate can elevate ROS production via the mitochondrial ETC and facilitate the assembly of NOX by phosphorylating p47*^phox^* [79]. Furthermore, ROS-induced damaged mitochondria are marked for clearance via mitophagy [80]. In parallel, palmitate contributes toward IR by downregulating canonical AKT signaling [81]. Elevated mitochondrial fission is a preceding event to mitophagy [82,83]. Thus, targeting mitophagy has been speculated as a means to alleviate skeletal muscle IR.

### 3.2. Molecular Pathways Involved in Mitophagy: A Novel Role for STX4?

One of the early responses to oxidative stress is the induction of mitophagy, a specialized form of autophagy that selectively disposes of aging, damaged or dysfunctional mitochondria (reviewed in [84]). Below we detail the canonical and non-canonical mitophagy pathways (Figure 4) [85,86,87,88,89,90,91], and integrate the emerging novel putative role for the exocytosis protein, Syntaxin 4 (STX4) in skeletal muscle IR-related mitophagy [76].

Canonical PARKIN-dependent mitophagy: The most defined mechanism of mitophagy is the ubiquitin-dependent phosphatase and tense homologue (PTEN)-induced putative kinase 1 (PINK1)-Parkin RBR E3 ubiquitin-protein ligase (PARKIN) pathway (Figure 4a). In healthy mitochondria, the cytosolic PINK1 serine/threonine kinase that translocates towards the mitochondria is imported across the outer- and inner-mitochondrial membranes (OMM and IMM) through their respective translocases (translocase of the outer membrane [TOM] and translocase of the inner membrane [TIM] complexes) prior to cleavage by presenilin-associated rhomboid like (PARL) proteases [92,93], resulting in a low basal level of PINK1 around healthy mitochondria. However, upon mitochondrial damage, the decreased membrane potential stabilizes OMM-localized PINK1, triggering autophosphorylation of Ser-228 and Ser-402 sites for PINK1 activation [89,94]. Activated OMM-localized PINK1 promotes PARKIN recruitment to the damaged mitochondria via phosphorylation of Ser-65 on OMM protein-associated ubiquitin [86,87,88,95]. Ser-65 phosphorylation of the recruited PARKIN activates its E3-ubiquitin ligase activity triggering a feedback loop of further Ser-65 phosphorylation on poly-ubiquitinated chains of OMM proteins, enabling more translocation of PARKIN proteins towards the mitochondria [86,87,88,95]. This sequence culminates in the recruitment of LC3-positive autophagosomes via autophagy receptors including p62 [85] and optineurin [96,97]).

Non-canonical receptor-dependent mitophagy: On the other hand, the receptor-dependent form of mitophagy does not rely on Ser-65 phosphorylated polyubiquitinated chains. Instead, OMM- and IMM-anchored proteins in possession of LC3-interacting motifs recruit LC3-positive autophagosomes by direct interaction (Figure 4b) [98,99,100]. To date, the most defined examples studied in mammalian cells include BCL2/adenovirus E1B 19 kDa protein-interacting protein 3-like (BNIP3L/NIX) [90,101] and FUN14 domain containing 1 (FUNDC1) [91,98,102,103] and most recently prohibitin-2, an IMM protein with a LC3-interacting motif that is involved in both ubiquitin- and receptor-dependent mitophagy [100,104].

Is STX4 a novel mitophagy associated protein?: STX4, is classically known as a key regulator of insulin-dependent glucose uptake by skeletal muscle, via its role as a SNARE protein at the sarcolemmal and t-tubule membrane surfaces [105,106]. We recently highlighted a novel localization for STX4 at the OMM, as well as a distinct mitochondrial role for STX4 at that locale [76]. Skeletal muscle-specific STX4 enrichment in HFD-fed mice led to reversal of IR, enhanced spontaneous activity and respiratory exchange ratio, as well as improvements in mitochondrial respiration and dynamics [76]. STX4 was found in the skeletal muscle mitochondrial fraction, and STX4 enrichment reduced skeletal muscle mitochondrial fragmentation and deactivated Drp1 in an AMPK-dependent manner [76]. Although this study focused on the role of STX4 in mitochondrial dynamics, studies have shown that mitochondrial fission proteins are also known regulators of mitophagy. For example, Drp1 within the mammalian brain, heart and pluripotent stem cells [107,108], and most recently the IMM fission protein MTFP1, have been identified as being involved in both PINK1-PARKIN and receptor-dependent mitophagy in human oral cancer cell lines [109].

Taken together, future studies focused on mitochondrial membrane proteins such as the recently discovered OMM-associated STX4 [76] may reveal a new player in mitophagy as it remains to be determined whether the absence of fragmented mitochondria in both lean and HFD skeletal muscle in STX4-enriched mice was due to increased repair and/or mitophagy for their clearance [76].

### 3.3. Mitophagy in Skeletal Muscle IR: Helpful or Harmful?

Although mitophagy is a key regulatory process for the maintenance of cellular homeostasis, in the diabetes field, a debate lingers on whether targeting mitophagy is beneficial for treating IR in prediabetes and T2D. Most pre-clinical in vivo studies focused on HFD-induced IR have evaluated PINK1/PARKIN-dependent mitophagy. For short-term HFD studies of 1–2 weeks [110,111] in mice, acute IR development corresponded with PINK1/PARKIN elevations in the skeletal muscle. In contrast, long-term HFD for 12–24 weeks in rodents [111,112] or 2 years in Rhesus monkeys [113] presented reduced expression of PINK1/PARKIN in the skeletal muscle. Despite the associative patterns of mitophagy with stages of IR across the pre-clinical models, these examples are limited by not distinguishing whether they represent the clinical characteristics of prediabetes or T2D. Nevertheless, one clinical study that focused on peripheral blood mononuclear cells reported that initial elevations of PARKIN/PINK1 and NIX mitophagy markers in prediabetes patients eventually decreased across newly diagnosed and advanced duration T2D patients [114]. Even though this study was not performed in insulin-sensitive tissues, the reported reduction of PINK1 levels in obese T2D skeletal muscle and progressive downregulation of mitochondrial protein in skeletal muscle from non-diabetes, to prediabetes and then T2D patients [115], could support a similar mitophagy model for diabetes. In this model, it is suggested that elevations of mitophagy associated with prediabetes and acute IR stages actually serve a protective role of mitigating accumulating mitochondrial damage (Figure 4c). However, as IR worsens towards T2D, mitophagy activity is impaired leading to excessive mROS production. Whether the decline of mitophagy is a cause or consequence of prediabetes progression into T2D remains to be determined. Although insulin-sensitizing lifestyle interventions of exercise and dietary restrictions [116] have been associated with recovering PINK1 and PARKIN in HFD-exposed IR models, emerging pre-clinical studies with flavonoids such as Bavachin and corylifol A in db/db mice [117], and isoflavonoid puerarin in palmitate-exposed L6 myotubes [118] also demonstrated improved insulin sensitivity with restored PARKIN/PINK1 activity. Although further therapeutic studies in the context of prediabetes and T2D are required, the evidence so far suggests that boosting mitophagy in the prediabetes stage could protect against accumulating mitochondrial damage and may serve as a useful preventative treatment.

Although evidence suggests that non-canonical receptor-mediated mitophagy may play a role in diabetes, the results have been quite mixed. Global FUNDC1-KO mice exhibited worsened HFD-induced IR [102]. Counterintuitively, skeletal muscle-specific FUNDC1-KO mice do not show obesity or IR when exposed to a 3-month-HFD [103]. Intriguingly, this was a result of skeletal muscle derived fibroblast growth factor 21 (FGF21) triggering thermogenesis in intercalating white adipose tissue via skeletal muscle-adipose cross talk [103]. Further, pancreatic-specific FUNDC1 overexpression also protected mice from a 4-month-HFD [91], suggesting that the requirement for FUNDC1 in modulating mitochondrial activity in response to IR may be tissue-dependent. A 3–12-week-HFD was associated with elevated BNIP3L/NIX in rats [119], whilst a later study investigating BNIP3L/NIX in C2C12 and primary human myotubes linked increased BNIP3L expression with impaired insulin signaling via mechanistic target of rapamycin kinase (mTOR)-dependent inhibition of insulin receptor substrate 1 (IRS-1) [101]. To date, no study has directly compared the differences between the PARKIN-dependent and -independent mitophagy pathways and their beneficial effects in prediabetes and T2D, and future studies are required to address this gap. PARKIN-independent mitophagy in T2D may be a tissue-specific requirement, and thus, an alternate model to the ubiquitin-dependent mitophagy model (Figure 4c).

Finally, it should be acknowledged that most of these mitophagy studies have been investigated under the obese and IR model of T2D (Figure 1). Interestingly, a recent 2023 study that measured PINK1 mRNA levels in the whole blood samples of obese-IR versus lean-T2D patients reported that decreased PINK1 was associated only with the obese cohort [120], suggesting that mitophagy may be different across the heterogenous T2D groups. Thus, the currently proposed mitophagy model (Figure 4c) may not be universally applicable to all T2D cohorts (Figure 1), and future studies need to investigate the impact of mitophagy in lean-T2D patients.

## 4. ROS-Related Molecular Pathways Underlying IR and T2D

T2D individuals show a deficit in skeletal muscle for the serine/threonine p21 (Cdc42/Rac1) activated kinase 1 (PAK1) [121], which was recently implicated in mitochondrial and glucose dyshomeostasis [121]. Muscle deficiency for PAK1 showed reduced expression of PGC1α [122], consistent with an essential role for PAK1 in maintaining redox balance via electron transport chain proteins in β-cells [123]. Several lines of evidence suggest that mROS-induced oxidative stress in skeletal muscle IR and T2D leads to maladaptive enhanced activation of the polyol pathway flux and advanced glycated end products (AGEs) and hexosamine pathways [124,125] (Figure 5). Further, hyperglycemia enhances polyol and hexosamine pathways leading to increased endogenous AGEs production (reviewed in [126]).

Polyol pathway: Chronic hyperglycemia in prediabetes and T2D induces the aldose reductase (AR)-dependent polyol pathway, which increases O_2_^•−^ generation and alters glucose and lipid metabolism thereby contributing to mitochondrial damage. Typically, AR reduces the toxic aldehydes to alcohol, but in a chronically hyperglycemic environment, it reduces glucose to sorbitol at the expense of NADPH. Additionally, in T2D, the hexokinase pathway becomes saturated, and the decrease in cytosolic free NAD^+^/NADH^−^ ratio changes the redox potential and enhances diacylglycerol (DAG) and triacylglycerol synthesis [127,128,129]. The overall reduction in NADPH levels reduces the regeneration of reduced glutathione (GSH) from its oxidized form. Further modeling this, the skeletal muscle of young male Wistar rats fed a HFD for 14 days showed reduced glutathione (GSH) and disulfide (GSSG) glutathione levels in the soleus muscle, as compared with chow-fed controls [130]. Further, MKR mice show reduced physical capacity, strength, and muscle mass leading to diabetic myopathy and increasing the risk of liver disease due to polyol accumulation [124]. In a recent clinical trial testing the effect of oral glutathione (GSH) supplementation for 3 weeks in obese subjects with and without T2D, the whole-body insulin sensitivity and skeletal muscle GSH significantly increased in the GSH group; oddly however, no changes were seen in skeletal muscle mitochondrial H_2_O_2_ emission rate, a measure of oxidative stress, pre- and post-GSH supplementation [131]. Thus, while it was concluded that oral GSH supplementation improved insulin sensitivity in obese patients without and with T2D, the link to skeletal muscle oxidative stress remained unclear.

Hexosamine pathway: IR-induced elevated fatty acid oxidation enhances fructose 6-phosphate levels that in turn activates hexosamine pathway and contributes to skeletal muscle IR. The rate limiting enzyme of this pathway is glutamine: fructose 6-phosphate amidotransferase (GFAT), which converts fructose 6-phosphate to glucosamine 6-phosphate and glutamate; upon further metabolism, this hexosamine biosynthesis pathway yields essential substrates for glycosylation of proteins and lipids. Additional major products of this pathway that accumulate in cells include UDP-*N*-Acetylglucosamine (UDP-GlcNAc) and UDP-N-acetylgalactosamine (UDP-GalNAc). Importantly, increased flux through the hexosamine pathway impairs glucose transport, contributing to the development of muscle IR [132,133]. Hexosamine pathway activity was reported to cause insulin resistance in C57BL/6NJ mice fed with HFD, and was related to molecular changes such as plasma membrane cholesterol accumulation and loss of filamentous actin, which is required for GLUT4 vesicle translocation and glucose uptake [134,135,136]. GLUT4 is *N*-glycosylated, and the absence of this modification prevents its subcellular localization during insulin stimulation and elevated hexosamine pathway flux could affect the *N*-glycan branching of glucose transporters and affect glucose uptake [137]. Moreover, individuals with T2D were found to have high GFAT1 activity determined from skeletal muscle biopsies, and GFAT activity in patients is related to postprandial hyperglycemia, oxidative stress, and other diabetic complications [138,139].

AGEs: Mitochondrial dysfunction is increased by AGEs [140]. AGEs are lethal metabolites that elicit oxidative stress via the receptor for advanced glycation end products (RAGE). RAGE activation is reported in high levels in chronic hyperglycemia [141,142,143,144]. Intracellular accumulation of AGEs alters protein structures and elicits oxidative stress. AGEs were initially implicated in skeletal muscle dysfunction leading to gradual and generalized loss of muscle strength [145]. More recently, AGEs are highly associated with sarcopenia in T2D individuals, resulting in decreased grip strength, knee extension strength and breakdown of muscular proteins [146]. In rats, AGE-albumin administration induced IR decreased GLUT4 expression and increased levels of nuclear factor NF-κB p50 subunit (NF-κB1) in skeletal muscle. Taken together, agents that reduce the development of AGEs would be predicted to prevent mitochondrial dysfunction and oxidative stress-induced IR. However, AGEs are broadly expressed and are a muscle-selective-drug targeting challenge. Nevertheless, AGEs are under scrutiny as biomarkers for earlier detection of diabetes-risk in bio-accessible fluids.

## 5. Mitochondrial Dysfunction and Islet β-Cell Dysfunction/Failure

Chronic overnutrition leads to a hyperactive metabolic state in β-cells, and the subsequent metabolic stress is related to suprathreshold ROS levels in β-cells. High glucose (glucotoxicity) and lipids (lipotoxicity) are known to cause β-cell damage via ROS, resulting in impaired GSIS (reviewed in [147]), mitochondrial defects and epigenetic modifications [148]. Reduced mtDNA content in β-cells is a marker of oxidative stress [149]. Beta cells are particularly susceptible to oxidative stress due to their high metabolic rate combined with low levels of antioxidant enzymes [150,151]. Thus, any impairment in the pro-oxidant/anti-oxidant balance ultimately leads to the development of T2D (reviewed in [152]). A thioredoxin/reductase-dependent mechanism enables β-cells to acquire the capacity to combat micromolar levels of H_2_O_2_ (50 µM), whereas a 100 µM bolus dose of H_2_O_2_ reduces β-cell viability and induces DNA damage, suggesting that the threshold limit for β-cells to handle the pro-oxidant capacity is less than 100 µM [57,153]. It is appropriate to point out that lean normoglycemic progeny from T2D patients have increased serum levels of 8-OHdG, an oxidative stress marker [154]. This begs the question whether oxidative stress is present in genetically predisposed subjects and induces the β-cell dysfunction in lean subjects in the absence of IR. Additionally, a number of chromosome 21q genes that contribute to T2D in lean individuals were identified and *KCNJ15* (potassium inwardly-rectifying channel, subfamily J), a known T2D susceptibility gene, was among them [155]. T2D human islets and in vitro exposure of high glucose demonstrated elevated levels of KCNJ15; whether this elevation is a cause or consequence of T2D is yet to be determined.

### 5.1. Abnormal Mitochondrial Function in PAK1-Deficient β-Cells

The molecular regulators of mitochondrial function in β-cells are just beginning to be elucidated. Islets from T2D donors show an 80% deficiency for PAK1 compared to control individuals, suggesting that PAK1 deficiency might contribute to T2D pathogenesis [156,157]. We recently reported a vital requirement for PAK1 in modulating ROS in β-cells via its interaction with mitochondrial NADH:ubiquinone oxidoreductase subunit A12 (NDUFA12) [123], a protein that is critically required for oxidative stress regulation [158] (Figure 3). Beta cells deficient for PAK1 exhibited decreased mitochondrial ETC proteins (CI, CII and CIV), impaired mitochondrial respiration, an imbalance in redox homeostasis, reduced GSIS, and reduced β-cell survival in human and rodent β-cells. PAK1 is critical for the formation of the mitochondrial ETC super complexes (CI-CIII and CI-CIII-CIV) and mitochondrial ETC function by binding to and regulating NDUFA12, the CI assembly and stabilizing subunit. Given that PAK1 protein plays two distinct roles, as a kinase, and as a protein scaffold, further investigations will require evaluation of these roles independently, using kinase dead mutants and kinase inactivating compounds, such as IPA3. Further understanding the PAK1 select signaling and scaffolding requirements will inform the downstream pathways that promote mitochondrial function and β-cell survival [123].

### 5.2. Abnormal Islet β-Cell Mitochondrial Morphology and Dynamics

Abnormal mitochondrial morphology is observed in cadaveric T2D donor β-cells [159]. Distorted mitochondria appear morphologically round and swollen, which impacts mitochondrial function, culminating in impaired GSIS [160,161,162]. Beta cells from T2D and non-diabetic human donors possess similar mitochondrial numbers but T2D donors exhibit significantly higher mitochondrial volume density [159,163]). Several defects in the activation of key enzymes that link glycolysis with tricarboxylic acid cycle have been identified. Each mitochondrion exhibits an increased workload in the T2D milieu, leading to high membrane potential, ROS formation and reduced mtDNA copy number(reviewed in [163]. Under T2D milieu conditions, β-cell mitochondria lose their ability to undergo fusion and become fragmented, leading to β-cell apoptosis [164].

In Goto-Kakizaki (GK) rats, a genetic-lean rat model that exhibits a prediabetes phase similar to humans prior to T2D onset, pancreatic islet β-cells showed mitochondrial network disintegration [165]. In addition, Zucker diabetic fatty rats and HFD-induced models of prediabetes showed mitochondrial remodeling, altered cristae structure, and increased β-cell apoptosis [166,167]. In a model of lean T2D, the Cohen diabetic sensitive (CD) rat, a hyperglycemic model, exhibited markedly diminished β-cell mitochondrial COX activity, defective mitochondria, and increased ROS levels without IR [168,169,170].

### 5.3. Mitophagy and Elevating Islet β-Cell Dysfunction

Emerging studies of dysregulatory metabolic states representative of prediabetes and T2D are identifying modulation of mitophagy as having a beneficial impact of mitigating disease progression in tissues such as pancreatic β-cells. Recently, it was suggested that mitophagy-deficient β-cells were susceptible to inflammatory stress and overexpressing the mitophagy regulator CLEC16A could protect β-cells from stress [171]. Natural antioxidant compounds have been known to modulate mitophagy and restore mitochondrial dynamics in T2D. However, the efficacy of mitophagy induction by these agents is limited and future studies exploring new avenues are needed, given the fact that mitochondrial dynamics are also important for normal β-cell function. Restoring canonical PARKIN-mediated mitophagy was shown to improve β-cell survival [91,172]. Furthermore, beneficial therapeutic effects in β-cells were observed with selenium nanodots, a synthetic prodrug of antioxidant glutathione peroxidase [173], suggesting the utility of restoring mitophagy to ameliorate oxidative stress in T2D models. While it is still under debate as to whether elevating mitophagy is the appropriate strategy to counteract chronic oxidative stress damage, there is consensus regarding the concept of balancing mitophagy and mitochondrial biogenesis to restore β-cell function.

### 5.4. Mitochondrial-Endoplasmic Reticulum Miscommunication in Islet β-Cell Dysfunction

Recent evidence demonstrated an association between ER-mitochondrial miscommunication and β-cell dysfunction in individuals with T2D (reviewed in [174,175]. Mitochondria and ER are in close proximity in the β-cell, and they interact via protein bridges and are highly dynamic, forming specific microdomains termed mitochondria-associated membranes (MAM) (reviewed in [176]. It is well established that MAM play a key role in cellular Ca^2+^ homeostasis [177]. MAM are implicated in regulating mitochondrial dynamics and function, oxidative metabolism, and apoptosis (reviewed in [178]). Furthermore, induction of ER stress by toxic lipids such as palmitate significantly reduced ER-mitochondria crosstalk and altered GSIS in the murine MIN6-B1 β-cell line [175]. Additionally, alterations in ER-mitochondrial crosstalk disrupts lipid homeostasis and is associated with T2D pathogenesis [175]. MAMs can also contain enzymes required for cholesterol and ceramide biosynthesis; there are several reports thus far suggesting elevated ceramide in T2D human islets [166,179]. Ceramide significantly alters the mitochondrial membrane potential (Δψ_M_), which can trigger the release of proapoptotic factors like cytochrome *C*, which activates pathways that culminate in β-cell apoptosis [180]. Disruption to MAM assembly compromises mitochondrial dynamics and bioenergetics, and promotes ROS production resulting in the release of mitochondrial-derived proapoptotic factors that promote β-cell death (reviewed in [178]).

We recently revealed that PAK1 enrichment in T2D human islets attenuated markers of ER stress—phosphorylated eukaryotic initiation factor-2α (eIF2α) and C/EBP Homologous Protein (CHOP), and improved β-cell GSIS function [123]. Given also the requirement for PAK1 in β-cell mitochondrial function [123], it remains to be determined if PAK1 functions in the MAM. Taken together, MAM plays a crucial role in dysfunctional β-cells and targeting MAM could be a potential interventional strategy for T2D.

## 6. Inter-Organ Miscommunications and Oxidative Stress in Insulin Resistance

Conditioned medium experiments and parabiosis experiments have revealed the presence of factors that facilitate inter-tissue/organ communication with other organs/tissues. For example, we recently discovered factor(s) released by skeletal muscle into conditioned medium, that when applied to β-cells, can significantly boost islet β-cell function [121]. Moreover, factors can be shed as soluble factors, such as many hormones (e.g., insulin), and it is now widely appreciated that the vast variety of factors can be shed encased in nanoscale structures.

Cells continuously shed extracellular vesicles (EVs), which are nanoscale bi-lipid structures (~35–1000 nm) that carry biomoleculer cargo, lack a functional nucleus and replicative capacity [181]. EVs have the potential to play a pivotal role as mediators of IR and pancreatic β-cell mass failure in T2D [182,183]; elevated glucose levels, inflammation, increased fatty acids, all of which contribute to oxidative stress, can lead to alterations in the quantity and composition of EVs released from the skeletal muscle, adipose, and pancreas (reviewed in [184]). This section will highlight the emerging role of EVs and a new class of mitochondria-containing EVs (Mito-EVs) [181] in promoting inter-tissue miscommunication during prediabetes and T2D.

### 6.1. Skeletal Muscle-Derived EVs-Islet β-Cell Crosstalk

Skeletal muscle-derived EVs, possibly containing myokines and microRNAs (miRNAs) [185,186]), have the intriguing ability to transfer lipid-induced IR between muscle cells and β-cells [187,188]. Further, in vivo, EVs from lipid-induced IR muscles can be incorporated into recipient pancreatic β-cells and affect changes to gene expression in those β-cells [187]. Muscle contraction-induced myokines were recently shown to influence glucose uptake, insulin sensitivity and fat metabolism (reviewed in [189]). Furthermore, these secreted myokines from contracted skeletal muscle, including C-X-C motif chemokine ligand 10 (CXCL10), irisin, chemokine ligand 1 (CX3CL1), follistatin and miR-133a (Table 1) also impact the function and survival of β-cells (Figure 6). Thus far, there is potential for skeletal muscle-derived miR-133a to target β-cell mitochondrial UCP2 and influence insulin secretion, given that an increase in skeletal muscle miR-133a leads to reduced UCP2 levels [190] and that β-cell UCP2 controls ROS production to promote GSIS [191]. Indeed, a marked rise in plasma EV protein content, especially antioxidant protein cargo (superoxide dismutase, catalase, and peroxiredoxin) is observed post-exercise to maintain a balanced redox state [192,193]. Currently it is unknown if EV-containing myokines influence pancreatic β-cell mitochondrial function and morphology within the context of prediabetes.

### 6.2. Adipose-Islet β-Cell Crosstalk and Mitochondrial Transfer

Recently, β-cells and adipose tissue were demonstrated to shed Mito-EVs under pathophysiological conditions (Table 2). There is potential for adipose-derived Mito-EVs to negatively influence β-cells, impairing their functionality in obesity-induced IR/T2D. For example, MIN6 β-cells exposed to EVs derived from mitochondrial stressed adipocytes obtained from caveolin-1 knockout mice, displayed significant cell death and dysfunction [206]. Although the content of these EVs was not examined for mitochondrial constituents, this provides proof of concept implicating inflamed adipocytes in performing detrimental horizontal transfer via EVs. By contrast, adipose-derived Mito-EVs determined to be carrying select mitochondrial proteins (PGM1, PCX and MDH1) positively impacted β-cells: β-cells exposed to Mito-EVs derived from epidymal white adipose tissue (eWAT) of diet-induced obese (DIO) mice displayed increased GSIS [207]. Moreover, in vivo intraperitoneal administration of these Mito-EVs, under fasting conditions, increased plasma insulin content and glucose tolerance in recipient mice, while blockade of EV production with GW4869 prevented glucose tolerance improvements [207]. Interestingly, β-cells exposed to Mito-EVs containing the mitochondrial matrix protein SLC25A5 shed by eWAT adipocytes from lean mice displayed loss of GSIS function [207]. Collectively, these examples point to the need for in-depth characterization of these Mito-EVs and their molecular cargo that carries the capacity to significantly impact β-cell function (Figure 7).

Recent findings also point to β-cells shedding EVs that may have relevance to mitochondrial function. A protein lacking in abundance in T2D tissues, DOC2B (Double C2 domain-containing protein B), was implicated in mitochondrial function and structure in cancer cells [208]; our preliminary findings indicate that β-cells package DOC2B into the lumen of secreted EVs, by virtue of DOC2B’s tandem C2 domains [209]. These roles and locales for DOC2B represent a departure from the classic role for DOC2B as an exocytosis regulatory protein localized to the cytosol. Intriguingly, this finding sparks the question as to whether DOC2B’s requirement for β-cell GSIS stems from a role in the mitochondria, and the potential of DOC2B-laden EVs to serve as indicators of mitochondrial status in β-cells. Advances in EV technologies have deeply improved our understanding of EV mechanisms. However, due to the inherent disadvantages in current EV isolation methods available, distinguishing EVs sources from a specific tissue remains largely a challenge due to the overlap in biophysical properties like size, density and morphology as well as heterogenous protein expression. Future studies may consider targeting β-cell specific proteins to gain insight into the molecular signatures of β-cell-derived EVs observed with mitochondrial dysfunction in T2D.

### 6.3. Adipose-Immune Cell Crosstalk and Mitochondrial Transfer

Chronic inflammation associated with hyperglycemia and obesity can contribute to increased oxidative stress [210]. Adipose-derived EVs, which includes the subset of Mito-EVs, are implicated in influencing monocyte differentiation into pro-inflammatory macrophages (reviewed in [211]). Adipocyte-to-macrophage mitochondrial transfer serves a regulatory role in systemic metabolic homeostasis [212], and dysregulations in this process may lead to obesity-induced IR/T2D. For example, EVs derived from ob/ob mouse visceral adipocytes (ob-Mito-EVs), whose cargo was partly constituted of mitochondrial proteins HSP70 and HSP90, influence the differentiation of monocytes into macrophages, carrying increased levels of proinflammatory cytokines IL-6 and TNF-α [205]. In vitro, macrophages preincubated with ob-Mito-EVs negatively impact insulin signaling and insulin-stimulated glucose uptake in myocytes compared to control (Figure 7) [205]. In vivo, ob-Mito-EVs led to the development of glucose intolerance and IR in control mice [205]. In contrast, adipose-derived stem-cell (ADSC)-derived EVs (ADSC-EVs) educate macrophages towards an anti-inflammatory polarized M2 macrophage fate by upregulating Arg-1 and IL-10, without significantly increasing pro-inflammatory M1-related iNOS, TNF-α, and IL-12 [213]. Indeed, HFD-obese mice treated with ADSC-EVs showed significantly improved glucose tolerance and insulin sensitivity [213]. One caveat though is that the cargo in these ADSC-EVs was not characterized for mitochondrial proteins per se. Nevertheless, these data suggest that the cargo within the adipocyte EVs influences monocyte/macrophage signaling in insulin-sensitive tissues (Figure 7).

## 7. Antioxidant Interventions for Prediabetes and T2D

Currently, a healthy diet consisting of abundant plant-based foods is a primary intervention for prediabetes individuals. Given the association between oxidative stress and T2D, a plethora of preclinical and clinical investigations into the therapeutic potential of natural antioxidants (i.e., polyphenols and vitamins) are currently underway to mitigate disease progression. Below we will focus on the leading natural antioxidant molecules that are substantially researched in clinical trials for T2D.

Curcumin: Curcumin is a natural polyphenol that is derived from the turmeric plant *Curcuma longa.* In vitro, curcumin inhibits ROS production and NF-κB activation in IR C2C12 myotubes, demonstrating antioxidant and anti-inflammatory effects with potential benefits for treating T2D [214]. However, clinical trials and meta-analyses have reported mixed results. One meta-analysis capturing 72% of clinical trials showed curcumin improving fasting blood glucose levels and hemoglobin A1c (HbA1c), whilst only three out of the five studies showed a decrease in Homeostatic Model Assessment for Insulin Resistance (HOMA-IR) [215]. However, curcumin treatment had beneficial impact on improving antioxidant capacity and alleviating oxidative stress. Furthermore, a chemically-modified curcuminoid showed improved insulin tolerance, fasting glycemia and alleviated oxidative stress in GK rats [216]. Thus, future studies utilizing synthetic derivatives of curcumin could maximize the therapeutic efficacy in T2D.

Resveratrol: Resveratrol is a polyphenol with free radical- and H_2_O_2_-scavenging capabilities that is naturally found in plant-based foods. IR rat myotubes studies indicated that resveratrol could restore glucose uptake and fatty acid oxidation via AMPK mediated pathways [217,218,219]. In addition, T2D GK non-obese rats receiving 20 mg/kg/day of resveratrol [220], and HFD-fed C57BL6/J male mice receiving 100 mg/kg/day of resveratrol [219], demonstrated improvements in IR and lipid metabolism and diminishments in muscle inflammation. In a placebo-controlled study, T2D patients receiving 800 mg/day resveratrol for 2 months showed reduced protein carbonyl content and white blood cell O_2_^•−^ levels and increased total thiol content, showing improved antioxidant capacity [221]. However, meta-analyses revealed mixed results for resveratrol safety and effectiveness in T2D [222,223,224] and thus, adequately powered randomized controlled trials are needed.

Vitamin C: Vitamin C is a naturally occurring antioxidant in citrus-based fruits. The recommended adult dietary intake of vitamin C ranges from 40 to 110 mg/day (reviewed in [225]. Vitamin C supplementation ranging from 500 to 1000 mg/day in T2D patients has demonstrated diminished production of ROS/RNS, improved antioxidant capacity, and improved glycemic control [225,226,227,228]. However, the high heterogeneity for glycemic control and small sample sizes [225] limit the conclusions that can be drawn regarding vitamin C treatment for T2D.

Vitamin D: Vitamin D is a lipid-soluble secosteroid; vitamin D_3_ is the animal-derived version formed from 7-dehydrocholsterol in the skin from natural sunlight. Vitamin D functions in genomic regulation via its receptor, and regulates signaling pathways that include MAP kinase and protein kinase C. In skeletal muscle, vitamin D treatment is associated with improved insulin sensitivity, reversal of myosteatosis, and decreased inflammation activation via NF-κB in HFD-fed mice [229]. Oxidative stress is inhibited in vitamin D-treated C2C12 muscle cells [230], suggesting a therapeutic role for vitamin D in T2D. A meta-analysis revealed that T2D patients receiving vitamin D supplementation to correct a vitamin D deficiency had reductions in HbA1c levels, fasting glucose and HOMA-IR [231]. Furthermore, placebo-controlled trials showed that vitamin D supplementation (4000 IU per day) lowers the risk of T2D progression in prediabetic individuals [232]. High doses of vitamin D at 100,000 or 200,000 IU biweekly over 6 months was associated with a reduction in oxidative stress [233]. Vitamin D supplementation may also be useful at decreasing inflammatory markers MCP-1 and IL-8 and attenuating oxidative stress in T2D [234]. Intriguingly, other natural antioxidant therapeutic candidates (e.g., zinc, omega-3 fatty acids, and vitamin E) have not shown any impact on glycemic control suggesting that vitamin D may be a leading antioxidant candidate for T2D [235].

## 8. Perspectives: Mitochondria and ROS as Therapeutic Targets for Prediabetes

One long-standing debate within the metabolic research field is whether mitochondrial dysfunction contributes to skeletal muscle IR, or is merely a bystander. A recent study that advocated the bystander explanation reported no difference in mitochondrial dysfunction between skeletal muscle biopsies from non-diabetic obese and T2D obese cohorts [236]. However, the non-diabetic obese cohort [236] had HbA1c levels that overlapped with the established prediabetic range of 5.7–6.4% [115]. In contrast, a recent proteomics study on skeletal muscle biopsies from T2D, prediabetic or non-obese human males identified decreases in mitochondrial proteins during prediabetes, which worsened upon progression to T2D [115]. These findings highlight the need to distinguish the mitochondrial activity patterns between T2D and prediabetes.

It is perhaps inadequate to solely focus on mitochondrial activity when assessing mitochondrial dysfunction. Evidence for subcellular mitochondrial heterogeneity (i.e., variance in mitochondrial ultrastructure and dynamics) is emerging in islet β-cells [237], and metabolic dysregulation can increase this heterogeneity [164]. Analyzing the degree to which mitochondrial heterogeneity contributes to skeletal muscle IR and β-cell dysfunction within the four T2D archetypes should further contribute to developing different therapeutic strategies for targeting mitochondrial dysfunction in prediabetes.

No drug treatments in the United States are currently approved for prediabetes, although off-label prescribing of metformin is recommended for select individuals with prediabetes [238]. Of relevance to the focus of this review, emerging evidences suggest that metformin reduces mitochondrial dysfunction in T2D individuals by reducing mROS production, preventing pro-fission mitochondrial morphology, boosting the antioxidant response, and inducing an increase in mitophagy [239,240]. Consistent with this, metformin reduces the mitochondrial pro-fission phenotype via AMPK in diabetic mice [241]. It will be beneficial to determine the prevalence of metformin use and the degree to which who is benefiting from metformin use in the prediabetes clinic setting given the heterogenous treatment effects reported for T2D [242]. Furthermore, there are serious side-effects associated with metformin use (e.g., lactic acidosis) and tolerability is individualistic. Thus, new safe and effective therapies that target novel mitochondrial dysfunction pathways and has a high benefit: risk ratio for prediabetes are needed.

Focusing on the prediabetes stage may also provide insights into the potential for effectiveness of antioxidant supplements at halting disease progression. T2D disease heterogeneity and the degree of oxidative stress are currently limiting factors in assessing antioxidant supplement efficacy to mitigate an individual’s oxidative stress. For example, three analyses indicate differential conclusions: (1) the Bialystock PLUS population identified a trend of lower total antioxidant capacity in individuals with prediabetes as compared to healthy controls in middle-aged groups [30]; (2) a meta-analysis encompassing 11 studies across Asia with different polyphenol, vitamin and mineral supplementation showed improvements with IR was correlated with total antioxidant capacity improvements [31]; (3) the 2019 Rotterdam study that evaluated antioxidant capacity using the nutritional gold standard of ferric reducing ability of plasma (FRAP) assay [28] across 5796 individuals identified that even though multi-antioxidant dietary rich intake lowered the risks of T2D per HOMA-IR scores, the risk of prediabetes was not reduced [29]. Furthermore, suboptimal dietary intakes (i.e., low antioxidant rich foods) were shown to be responsible for 7/10 newly diagnosed T2D cases across and within the populations of 184 high- and low-income countries [27]. Thus, the natural heterogeneity of access to antioxidant rich diets within various populations and geographic locations, as well as individual differences need to be taken into account when evaluating antioxidant therapeutic efficacy.

In summary, significant progress has been made thus far in the metabolic field. The body of evidence supports the notion that (1) T2D is heterogenous in nature and IR does not necessarily precede T2D development, (2) oxidative damage occurs early during disease onset, (3) boosting mitophagy could improve islet β-cell and skeletal muscle function, and (4) cargo shed from metabolically stressed tissues may contribute towards T2D progression. Further research is needed to understand and elucidate the regulation and relevance of ROS within the various subcellular microdomains in health and metabolic dysfunction. Measurement of ROS in the subcellular microdomain may help overcome some of the challenges faced when developing new therapies dealing with the heterogeneity that exists within T2D. Currently, individuals are stratified based on BMI for therapeutic interventions, yet due to phenotypic heterogeneity, this does not capture all in this “one-size-fits-all” approach. Future studies may consider targeting ROS and mitochondrial dysfunction, as a means to treat more T2D archetypes versus obese-T2D or lean-T2D. Future focus on developing synthetic derivatives of natural antioxidants, multi-faceted therapies that dually promote pancreatic and skeletal muscle glucose metabolism and mitochondrial function, expanding clinical studies to include a focus on prediabetes and taking individual antioxidant capacities into account may provide the best strategy for preventing T2D progression.

## Figures and Tables

**Figure 1 ijms-25-01504-f001:**
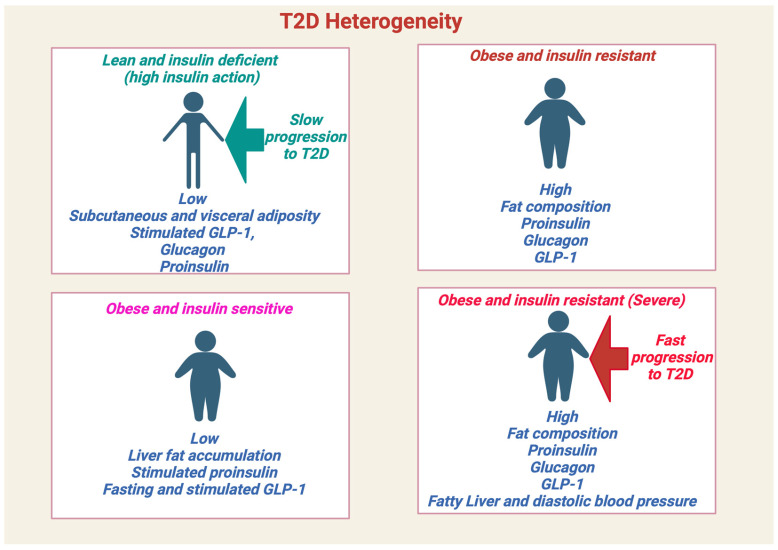
Phenotypic and Clinical manifestation of T2D heterogenous populations. Classification of T2D based upon specific characteristics features including fat accumulation, insulin action, β-cell function, multi-omics biomarkers and disease progression. Created with BioRender.com.

**Figure 2 ijms-25-01504-f002:**
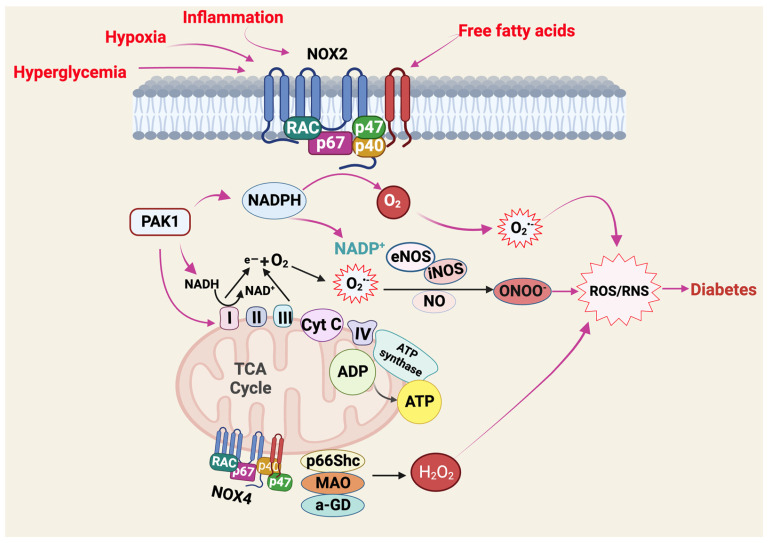
Mechanisms of chronic ROS/RNS generation in dysfunctional mitochondria. Exposure of cells to diabetogenic conditions like hypoxia, inflammation, hyperglycemia, and overproduction of free fatty acids can lead to increased leakage of electrons from the mitochondrial ETC. This enhanced electron leakage results in the overproduction of ROS/RNS, which can lead to oxidative stress and cellular damage. Created with BioRender.com.

**Figure 3 ijms-25-01504-f003:**
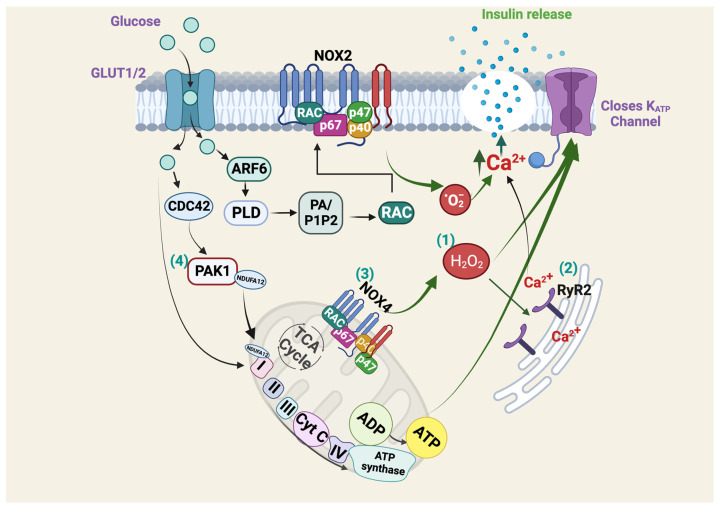
Transient ROS signaling in the healthy islet β-cell. (**1**) Glucose metabolism increases the levels of ROS in a transient manner, which triggers signaling events that ultimately culminating in GSIS. (**2**) The increased levels of ROS (H_2_O_2_) activate ryanodine receptors (RyRs) at the surface of endoplasmic reticulum, which causes the release of Ca^2+^ sufficient enough to trigger GSIS. (**3**) NOX4 at the mitochondrial surface causes an increase in ROS that causes the closure of ATP sensitive potassium channel and ensuing GSIS. (**4**) The plasma membrane associated NOX2 activation requires GTPase signaling cascades that lead to mobilization of insulin secretory granules towards the plasma membrane for release of insulin. Created with BioRender.com.

**Figure 4 ijms-25-01504-f004:**
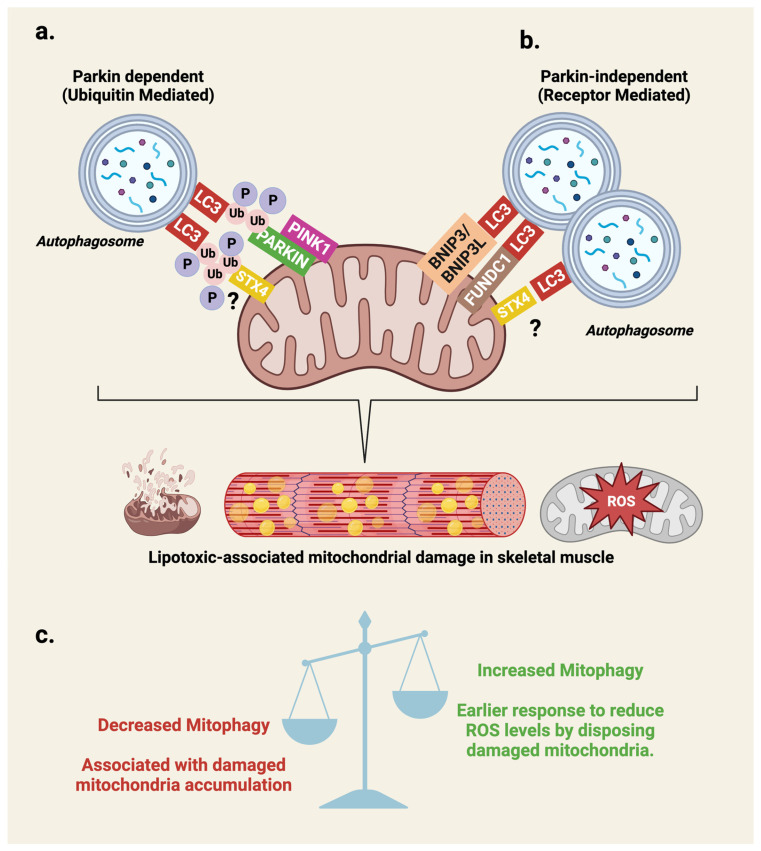
Differential mechanisms involved in the contribution of mitophagy associated with various stages of skeletal muscle IR. Mitophagy is a specialized form of macro-autophagy that selectively disposes of aging, damaged or dysfunctional mitochondria via autophagolysosomes-mediated degradation. (**a**) Ubiquitin-dependent mitophagy relies upon the stabilization of PINK1 on the OMM upon mitochondrial membrane damage and/or decrease of membrane potential. PINK1 autoactivation via Ser 228 and 405 autophosphorylation enables Ser 65 phosphorylation of ubiquitin on OMM proteins to recruit PARKIN. PINK1 activation of PARKIN via Ser 65 phosphorylation activates its E3-ubiquitin ligase activity triggering a feedback loop of further Ser 65 phosphorylation of polyubiquitin chains on OMM proteins and recruitment of more PARKIN. The ser-65 phosphorylated polyubiquitin chains of the OMM proteins facilitate LC3-positive autophagosome recruitment via LC3 interacting adaptor proteins. (**b**) Receptor-dependent mitophagy relies upon OMM and inner mitochondrial membrane (IMM) proteins with LC3-interacting motif regions such as FUNDC1 and BNIP3L/NIX that directly bind to LC3 enabling mitochondria-autophagosome fusion. (**c**) Proposed model for mitophagy role in T2D where increased mitophagy acts as a protective early response to protect against accumulative mitochondrial damage via ROS production, whilst T2D muscle the decreased mitophagy activity enables damaged mitochondria accumulation. Whether STX4 regulates mitophagy via a PARKIN-dependent or -independent pathway remains in question (?). Created with BioRender.com.

**Figure 5 ijms-25-01504-f005:**
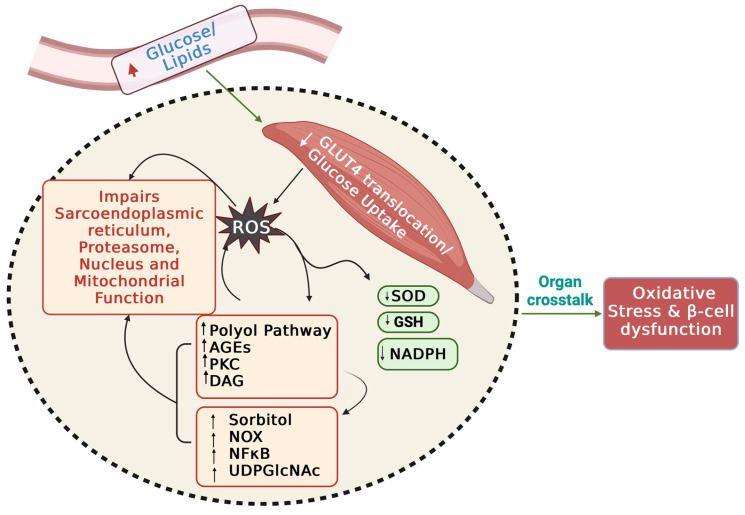
Mechanisms of skeletal muscle ROS generation in T2D. Hyperglycemia-mediated mitochondrial defects and the activation of stress pathways increase ROS in skeletal muscle, which then contributes to islet cell damage/dysfunction. GSH: glutathione; SOD: superoxide dismutase; NADPH: nicotinamide adenine dinucleotide phosphate; AGE: advanced glycation end products; PKC: protein kinase C; DAG: diacyl glycerol; NF-κB: nuclear factor-kappa-B; PKC; NOX: NADPH oxidases; UDPGlcNAc: uridine diphosphate-N-acetylglucosamine. The (↑) indicates upregulation/increase, and (↓) arrow indicates downregulation/decrease in the stress associated pathways, signaling molecules and antioxidants. All the connecting arrows specify the sequence of cellular responses induced by ROS via elevated levels of extra/intracellular glucose and lipids. Created with BioRender.com.

**Figure 6 ijms-25-01504-f006:**
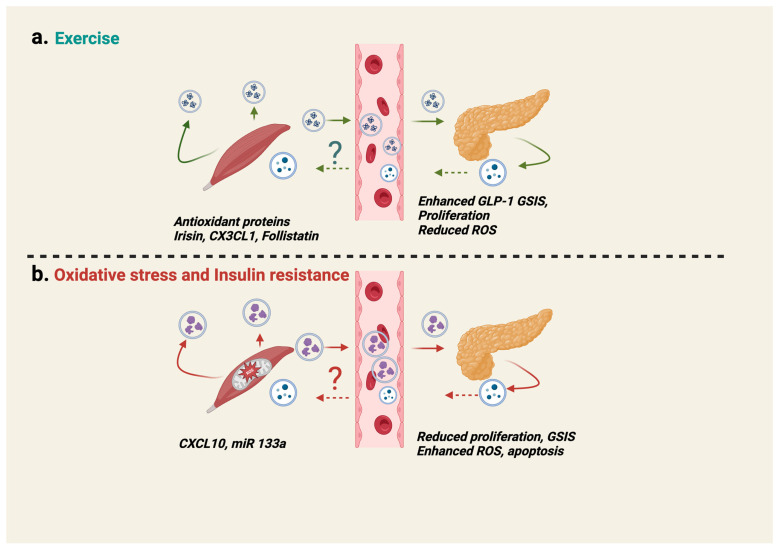
Islet β-cell-skeletal muscle crosstalk is mediated by EVs. During exercise (**a**) or IR (**b**), skeletal muscles secrete EVs which contain numerous secreted factors, including mitochondrial cargo. These factors can impact the health of islet β-cells. Refer to Table 2 for more details. GLP-1, Glucagon-like peptide-1. Solid arrows indicate the release of EVs from the tissues (skeletal muscle and islet β-cell). The dashed arrow and (?) indicates that it remains to be determined as to whether EVs released from an islet β-cell are required to regulate skeletal muscle function. Created with BioRender.com.

**Figure 7 ijms-25-01504-f007:**
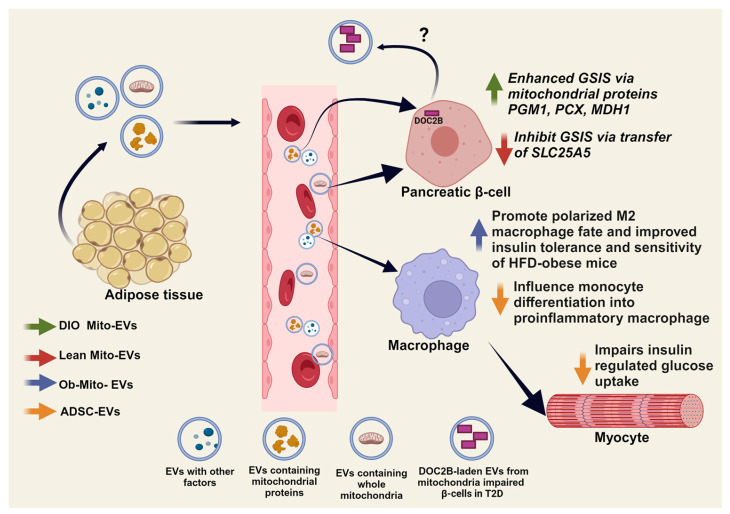
Adipose tissue Mito-EV mediated inter-organ communications. Mito-EV cargo impacts whole-body metabolism by influencing pancreatic β-cells and macrophage function. Negative effects (↓) on macrophage lead to impaired insulin stimulated glucose uptake by skeletal muscle myocytes and decreased GSIS from pancreatic β-cells. Positive effect (↑), increased insulin tolerance and sensitivity, or increased GSIS from pancreatic β-cells. Created with BioRender.com.

**Table 1 ijms-25-01504-t001:** Skeletal muscle-secreted factors, linked to oxidative stress (or ROS), insulin resistance, or Type 2 diabetic conditions and their effects on pancreatic β-cells.

Myokines or miRNA from Skeletal Muscle EVs	Relevance with Oxidative Stress (ROS), Insulin Resistance (IR) or Type 2 Diabetes (T2D)	Effect on Pancreatic β-Cells	Ref.
Follistatin	Reduces ROS in T2D	Enhances GSIS and β-cell proliferation	[194]
Irisin	Reduces inflammation-mediated oxidative stress	Enhances insulin production, GSIS and β-cell proliferation	[195]
CX3CL1	Reduces ROS	Enhances GSIS	[196,197]
miR-133a	Expression increased during oxidative stress	Reduces GSIS	[198]
CXCL10	Enhanced by oxidative stress or IR	Enhances apoptosis and reduces β-cell proliferation	[199]

**Table 2 ijms-25-01504-t002:** Mitochondrial cargo within Mito-EVs linked to oxidative stress, insulin resistance or Type 2 diabetic conditions and their effects on recipient cells.

Cell Source	Mitochondrial Cargo within Mito-EVs	Relevance to ROS, IR, and T2D	Effects on Recipient Cells	Ref.
β-cells	PHGDH, ALDH9A1, ADSL, and DNML1	Increase inflammation-mediated oxidative stress	Horizontal transfer of EVs promoted immune cell recruitment to islets insulin-positive β-cells	[200]
Adipocytes	Fatty acid oxidation enzymes ECHA and HCDH	Tumorigenicity exacerbated in obesity	Adipocyte-EVs promoted melanoma tumor cell progression via transfer of fatty acids under obesity conditions in vivo and ex vivo	[201]
Adipocytes	HSP60, VDAC1, COXIV	Promote ROS under obesity	Induce mitochondrial dysfunction of the host network and free radical production in cardiomyocytes. A beneficial effect that protects the heart from ischemia reperfusion injury	[202]
Adipocytes	mtDNA, AMP, ATP, that are MitoTracker Green (MTG) positive, without changes to PDHβ	Increased ROS	Adapt EVs as a quality control to remove damaged mitochondria	[203]
Adipose tissue MSCs and adipocytes	PGM1, PCX, MDH1	Insulin secretion under obesity-induced IR/T2D	Promote GSIS in β-cells	[204]
Adipocytes	HSP70 and HSP90	Promote IR in myocytes	Influence monocyte differentiation into pro-inflammatory macrophages, which directly act on myocytes to impair insulin-stimulated glucose uptake and insulin signaling in vitro and in vivo	[205]

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
