# Peer review of "Mitochondrial Dysfunction, Oxidative Stress, and Inter-Organ Miscommunications in T2D Progression"

_ijms, 2024, doi:10.3390/ijms25031504_

Round 1
Reviewer 1 Report
Comments and Suggestions for Authors
The manuscript aimed to review the roles of mitochondria, ROS and extracellular vesicles (EVs) in Type 2 diabetes progression. Four T2D archetypes were described, linking IR in the prediabetes stage. The effects of extracellular vesicles on islet beta-cells, originating from adipose tissue (Mito-EVs) and skeletal muscle-derived EVs, were described to illustrate inter-organ miscommunications. Antioxidants were discussed for their potential to control ROS and the progression of T2D. The manuscript addressed several interesting issues related to T2D progression. Readers need to digest to extract valuable information.
Major concerns:
1. The statement in Line 173 “Skeletal muscle account for 80% of the insulin-stimulated glucose disposal due to high metabolic demands” needs to be carefully addressed. In the original reference, it was “Glucose uptake by all tissues of the body reached 80% of its maximum value (12.6 mg/kg · min) at a plasma insulin concentration of ∼200 μU/ml.”
2. It is unclear how high fat acid (high fat diet) induce ROS production, insulin resistance (IR) and regulate mitophagy activity in the manuscript.
3. T2D and insulin resistance (IR) are different issues, as Figure 1 in this manuscript points out. The concept of insulin resistance is often used to link T2D in this article. However, there are many reasons and conditions contributing to T2D production. For example: 1) lean and insulin deficient, 2) obese and insulin sensitive, 3) obese and IR, and 4) a severe phenotype with obesity. T2D should be discussed separately rather than defining T2D progression only in terms of IR.
Minor concerns:
The title of Figure 2 and the content is not well-correlated, so as Figure 4.
Comments on the Quality of English LanguageN/A
Reviewer 2 Report
Comments and Suggestions for Authors
In this ms Veluthakal and colleagues review the literature describing mitochondrial dysfunction, oxidative stress and interorgan miscommunication in type 2 diabetes (T2D) and, more specifically, in the progression of T2D. Given the heterogeneous nature of T2D, the review considers the consequences of these pathomechanisms for insulin sensitivity as well as for insulin secretion (the role of glucagon and incretins is somewhat underreported). The overall purpose of the review appears to scan the literature for potential therapeutic targets to slow or even to halt the progression from pre-diabetes to overt T2D. In consequence the scope of the review is very broad, which impedes definite conclusions. The summary at the end states that while the importance of each of the above-named pathomechanisms is undisputed, nearly everything else is in dispute and more research is needed to settle the disputes. This leaves the reader who has worked through a quite densely packed load of information somewhat unsatisfied.
The authors emphasize the topicality of their review by stating that the heterogeneity ofT2D has emerged during the last decade. However, the polygenic inheritance of T2D (which underlies its heterogeneity) was known when I was a medical student (long ago). Similarly, the role of the insulin secretion deficit as independent contributor to T2D has been discussed for decades and has become generally accepted by the GWA data showing that the large majority of the T2D susceptibility genes affect beta-cell function. Finally, the phenotypic differentiation of T2D as presented in the IMI DIRECT study (Fig. 1), was preceded by the cluster analysis by Ahlqvist et al. and a comparison of how much the respective subgroups concur or differ would have been helpful.
In principle this review presents interesting information in a systematic way and the effort to bridge the gap, often existent between basic research and clinical research at is laudable. But this approach comes at a price: all too often the information is only superficially presented and for more insight the reader is referred to a review from which the information was taken. So the question comes up what the added value of the present accumulation of data and hypotheses is. A more convincing version may result when the authors consider which specific readership they want to address.
Round 2
Reviewer 2 Report
Comments and Suggestions for Authors
The revision has considerably improved the manuscript. I am still somewhat in doubt as to whether the review serves to critically evaluate strengths and weaknesses of current concepts of the pathogenesis of type 2 diabetes, but it clearly serves the purpose to give a tour d´horizon for a broad audience.